# Efficient YOLOv7-Drone: An Enhanced Object Detection Approach for Drone Aerial Imagery

**Xiaofeng Fu** [1,†], **Guoting Wei** [2,†], **Xia Yuan** [2,*], **Yongshun Liang** [3], **Yuming Bo** [1]

1   School of Automation, Nanjing University of Science and Technology, Nanjing 210094, China; fuxiaofeng@njust.edu.cn (X.F.); byming@njust.edu.cn (Y.B.)
2   School of Computer Science and Engineering, Nanjing University of Science and Technology, Nanjing 210094, China; weiguoting@njust.edu.cn
3   School of Mathematics and Statistics, Nanjing University of Science and Technology, Nanjing 210094, China; liangyongshun@njust.edu.cn
*   Correspondence: yuanxia@njust.edu.cn
†   These authors contributed equally to this work.

**Abstract:** In recent years, the rise of low-cost mini rotary-wing drone technology across diverse sectors has emphasized the crucial role of object detection within drone aerial imagery. Low-cost mini rotary-wing drones come with intrinsic limitations, especially in computational power. Drones come with intrinsic limitations, especially in resource availability. This context underscores an urgent need for solutions that synergize low latency, high precision, and computational efficiency. Previous methodologies have primarily depended on high-resolution images, leading to considerable computational burdens. To enhance the efficiency and accuracy of object detection in drone aerial images, and building on the YOLOv7, we propose the Efficient YOLOv7-Drone. Recognizing the common presence of small objects in aerial imagery, we eliminated the less efficient P5 detection head and incorporated the P2 detection head for increased precision in small object detection. To ensure efficient feature relay from the Backbone to the Neck, channels within the CBS module were optimized. To focus the model more on the foreground and reduce redundant computations, the TGM-CESC module was introduced, achieving the generation of pixel-level constrained sparse convolution masks. Furthermore, to mitigate potential data losses from sparse convolution, we embedded the head context-enhanced method (HCEM). Comprehensive evaluation using the VisDrone and UAVDT datasets demonstrated our model's efficacy and practical applicability. The Efficient Yolov7-Drone achieved state-of-the-art scores while ensuring real-time detection performance.

**Keywords:** object detection; drone aerial imagery; feature fusion; small object





## 1. Introduction

With their compact size and ease of operation, drones have become indispensable. They offer high flexibility and affordability. These devices are widely used in various sectors. They span from military operations to civilian tasks. Examples include disaster relief and traffic monitoring [1–3]. A critical technology supporting these diverse applications is object detection in drone-captured imagery.

In the field of object detection, deep neural networks have brought significant advancements. Their recent progress has markedly improved performance metrics. Notable benchmarks include MS COCO [4] and PASCAL VOC [5]. However, the majority of these deep CNNs were designed for natural scene images [6–8]. Drone-captured imagery is quite different from natural scene imagery. Thus, these networks often do not perform as well when applied to drone-captured content.

Designing object detectors for low-cost mini rotary-wing drone platforms is challenging. This challenge is distinct from conventional datasets.As shown in Figure 1, several challenges arise when adapting drone aerial images for object detection [9]:

- **Resource Limitations:** Low-cost mini rotary-wing drones have inherently limited computational capabilities. Specifically, these drones are constrained in terms of data processing and memory capacity. To equip these drones with real-time, high-precision object detection capabilities, there is an urgent need for a solution that ensures high accuracy and low latency while minimizing computational overhead.
- **Prevalenceof Small Objects:** Drone images often feature small, densely populated objects.
- **LimitedForeground Proportion:** The actual subjects of interest, or foreground objects, typically constitute a minor portion of the entire image.

Thesechallenges highlight the need for advanced, low-latency detection systems for drone imagery.

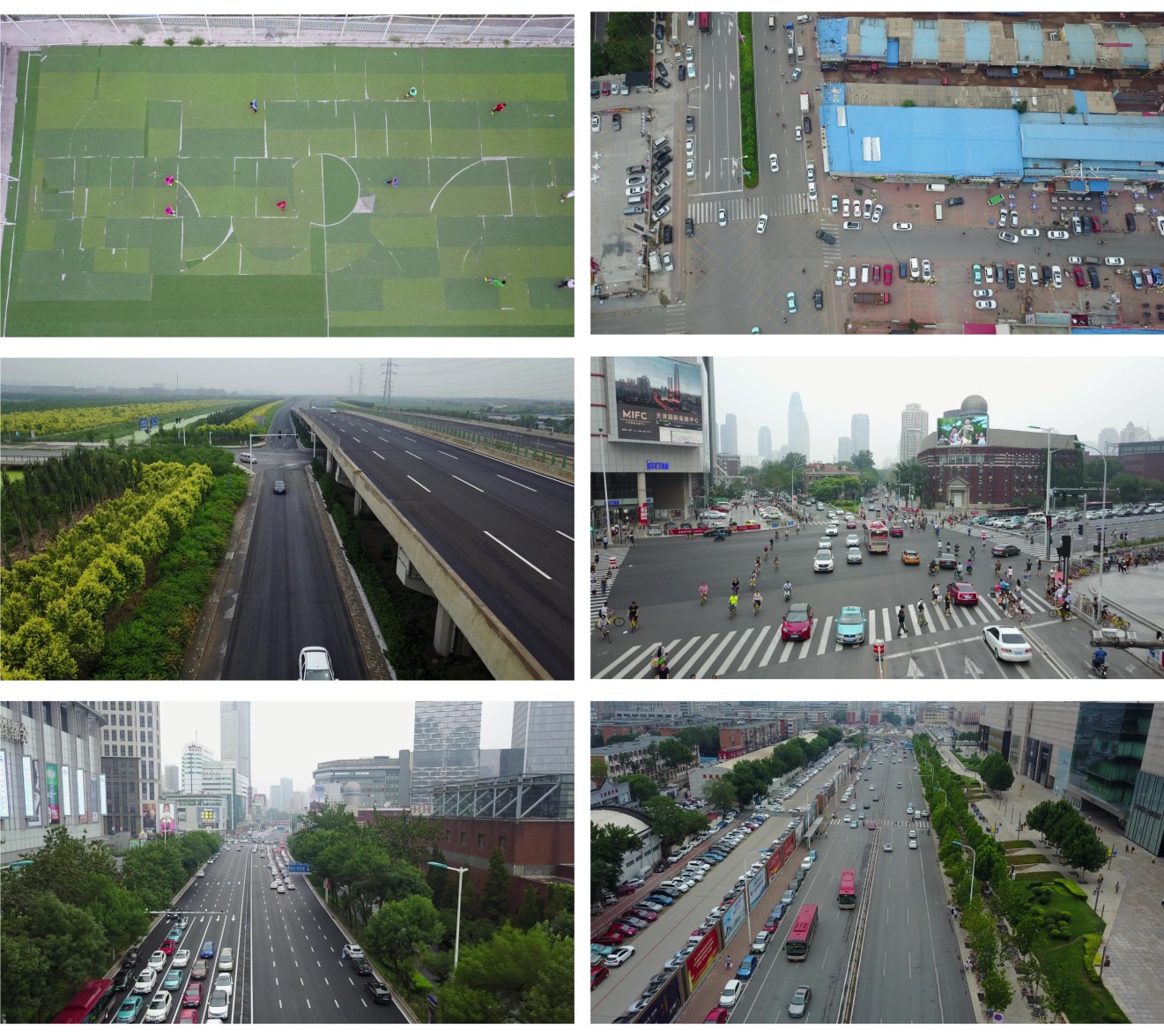

**Figure 1.** Visualization of the characteristics of drone-captured imagery. The first column illustrates the low proportion of foreground regions, while the second column emphasizes the prevalence of numerous and densely packed small objects.

Most research favors complex models to improve small object detection in aerial images. These models often rely on high-resolution inputs, consuming significant computational resources. This approach is misaligned with the inherent computational limitations of drone platforms. There is a highlighted need for efficient, lightweight models. However, complex object detection models offer high precision but are often unsuitable for edge device deployment due to their computational demands.In contrast, lightweight detectors might not maintain the same level of accuracy [10–12]. In an attempt to address this trade-off, numerous studies have focused on optimizing the primary network using methods such as network pruning [13,14] and structural redesign [15,16]. Though these methods prove effective for conventional images, their direct applicability to drone imagery remains questionable due to the distinct differences between conventional images and drone-captured scenes. Traditional detection strategies for aerial imagery often adopt a coarse-to-fine approach [17–20]. These methods use coarse detectors to discern larger instances and regions densely populated with smaller instances. Fine detectors are then applied to these highlighted regions for accurate identification of the small instances. While precise, the computational demands of these strategies make them less suitable for real-time drone applications.

Recognizing the limitations of existing object detection models in handling drone-captured images, we propose the Efficient YOLOv7-Drone, specifically designed to enhance object detection efficiency and accuracy in drone aerial imagery. Considering the dominance of smaller objects in aerial images and model efficiency, we removed the underperforming P5 detection head and introduced the P2 detection head, specifically to improve the detection of tiny objects. To ensure efficient feature relay from the Backbone to the Neck, we conducted channel optimization for the CBS module. Additionally, we made adjustments to other network components to boost the model's performance.

As the foreground often occupies only a small fraction of aerial images, the CEASC [21] module utilizes sparse convolution techniques [22,23], which narrows the network's attention, diminishing superfluous computations on background elements. It achieves this by generating a learnable mask. This ensures convolutions are performed only on select sparse sampling areas, optimizing computational efficiency. However, the performance of sparse convolution heavily depends on the quality of the generated mask. Traditional methods often employ fixed mask ratios to guide mask generation, presenting its own set of challenges. A mask ratio that is too small might lead to overly extensive sparse sampling regions, incurring unnecessary computations on the background and potentially compromising both efficiency and accuracy. Conversely, an excessively large ratio could shrink the sparse sampling areas too much. This risks omitting crucial foreground and contextual information, which in turn hampers detection performance. To address this, [21] introduced the adaptive multi-layer masking (AMM) scheme. It optimizes mask ratios across different feature pyramid network (FPN) levels using a custom loss function, thus balancing detection accuracy with efficiency. However, relying on the mask ratio to control mask generation could introduce significant uncertainties. Therefore, we proposed a novel module, TGM-CESC. Central to this approach is the target-guided mask approach, which leverages object labels to create foreground and background binary maps. By computing a particular loss with masks corresponding to sparse convolution, we achieved pixel-level guidance for generating sparse convolution masks.Subsequently, we integrated the TGM-CESC module into the Efficient YOLOv7-Drone, replacing the original re-parameterized convolution (RepConv) module.

To capture rich semantic information, we introduced the head context-enhanced method (HCEM). This method capitalizes on fusing feature maps from adjacent layers, compensating for the potential information loss at lower resolutions due to the mask quality in sparse convolution.

The main contributions of our work are as follows:

(1) The Efficient YOLOv7-Drone. In drone-captured images, we often observe objects that are small, densely packed, and of varied scales. To address these challenges, we present the Efficient YOLOv7-Drone. By omitting the P5 detection head, incorporating the P2 detection head, and fine-tuning the CBS module's channels, our model skillfully narrows the divide between performance and computational efficiency. This results in significant performance enhancements in object detection for drone-captured images.

(2) Target-Guided Mask Strategy. Recognizing the inherent sparsity of foreground elements in aerial images, we proposed the context-enhanced sparse convolution with target-guided masking(TGM-CESC) module. Central to this module is our target-guided mask strategy.By producing ground truth binary maps that correspond to the masks, we establish pixel-level constraints for generating sparse convolution masks. This offers an accurate and efficient solution for detecting sparsely scattered objects in aerial imagery, further sharpening detection precision amidst vast backgrounds.

(3) Head Context-Enhanced Method. To compensate for potential information loss induced by sparse convolution, we introduced the head context-enhanced method (HCEM). This strategy exploits the synergistic effect between feature map layers, merging features from adjacent levels, effectively countering the information loss due to the quality of masks in sparse convolution.

(4) We conducted comprehensive experiments on two popular datasets, VisDrone and UAVDT. The results decisively demonstrate that the methodologies we introduced for drone platforms can achieve real-time detection while maintaining a high level of accuracy.

## 2. Related Work

### 2.1. General Object Detection

Traditional object detection techniques, such as the scale-invariant feature transform (SIFT) [24], histogram of oriented gradients (HOG) [25], and deformable parts models (DPM) [26], predominantly employ the sliding window approach on images. Initially, these techniques recognize candidate regions within images, subsequently extract pertinent features, and employ the support vector machine (SVM) [27] classifier for categorization. Despite the accuracy of these conventional approaches, they grapple with issues of high computational complexity, sluggish processing speed, limited adaptability, and compromised robustness. However, given the swift evolution of machine and deep learning, deep-learning-driven object detection algorithms are increasingly overshadowing these traditional methodologies.

The general object detection methods based on deep learning are primarily concerned with natural images and can be broadly divided into multi-stage and single-stage detectors. Multi-stage detectors, like R-CNN [28], Faster-RCNN [29], Mask-RCNN [30], and Cascade R-CNN [31], begin with the generation of proposal regions through a region proposal network (RPN). Subsequently, objects within these proposed regions are classified and localized. On the other hand, single-stage detectors, such as the YOLO series [8,15,32–36], RetinaNet [7], GFL [37], and FCOS [38], classify and localize objects directly on the overall feature map, bypassing the step of using proposal regions. As a result, single-stage detectors are typically faster and are especially suitable for real-time object detection. With the rapid advancements in deep convolutional neural networks (DCNNs), the efficiency of single-stage detectors has also seen continuous improvement. However, while these methods have achieved commendable results in the general object detection domain, adapting them directly to UAV aerial image detection is not always ideal.

### 2.2. Object Detection in Aerial Images

Object detection in drone aerial imagery differs significantly from its counterpart in general imagery in three fundamental ways: (1) Aerial photographs predominantly feature small, densely packed objects. (2) Compared to general pictures, the proportion of the foreground in aerial images is relatively lower. (3) The resource limitations inher-

ent to drone platforms demand an optimized trade-off between detection accuracy and operational efficiency.

Much of the previous research has predominantly focused on enhancing the detection accuracy of small objects, with a coarse-to-fine strategy being commonly adopted. Initially, a coarse detector is designed to locate large-scale targets. For densely distributed small objects, methods such as clustering are employed to segment them into multiple sub-regions. Subsequently, a fine detector is used to further identify objects within these sub-regions. ClusDet [20] begins with a coarse detector to extract cluster proposals, then leverages the ScaleNet for refined detection. DMNet [19] uses density maps to fine-tune region selections. UFPMPDet [17] employs the Mosaic method to merge subregions identified by the coarse detector suppressing the background, and further introduces the multi-proxy detection network (MP-Det) to enhance small-object detection accuracy. HRDNet [39] combines image pyramids with feature pyramids to improve the accuracy of detecting small objects. However, the multi-staged nature of many such strategies means they are computationally intensive, rendering them sub-optimal for drone platforms constrained by resources.

### 2.3. YOLO Architectures Adapted to Aerial Imagery

In recent years, the YOLO algorithm has cemented its position as a frontrunner in single-stage object detection, receiving widespread recognition for its unmatched detection efficiency. Since the introduction of YOLOv4 [15], the YOLO series has consistently improved in accuracy and has started to outpace select two-stage detection techniques. Given the demands of drone-based object detection, where striking a balance between accuracy and efficiency is paramount, the YOLO algorithm emerges as the optimal choice. TPH-YOLOv5 [40], addressing the unique challenges of aerial imagery such as notable scale variations and high-density scenes, incorporates self-attention modules into its YOLOv5 prediction head. Furthermore, it integrates the CBAM attention module to bolster detection capabilities. Vit-YOLO [41] focuses on enhancing the detection of small objects by integrating a multi-head self-attention block and the BiFPN module. Building on this, TPH-YOLOv5++ [42] introduces the CA-Trans module, ensuring improved detection efficiency while maintaining performance levels. However, while these models achieve enhanced detection precision, the integration of transformer modules increases computational demands, leading to a significant drop in detection efficiency. This trade-off falls short of meeting the real-time detection demands of drone platforms. The latest in the YOLO series, YOLOv7 [36], brings notable enhancements in detection accuracy and computational efficiency. Its outstanding balance between precision and detection speed positions YOLOv7 as an optimal choice for drone aerial object detection. Thus, this research has selected YOLOv7 as the foundational model for further exploration and optimization.

### 3. Method

As illustrated in Figure 2, we specifically designed the Efficient YOLOv7-Drone model for drone-captured aerial imagery. This model incorporates the P2 detection layer. To achieve the optimal balance between detection accuracy and efficiency, we removed the underperforming P5 detection layer and adjusted the network structure accordingly. We introduced the TGM-CESC strategy to focus more on the foreground of aerial images. Finally, we integrated the TGM-CESC approach into the framework of the Efficient YOLOv7-Drone. Enhancing the framework further, we applied the head context-enhanced method (HCEM), ensuring both precision and efficiency in detection.

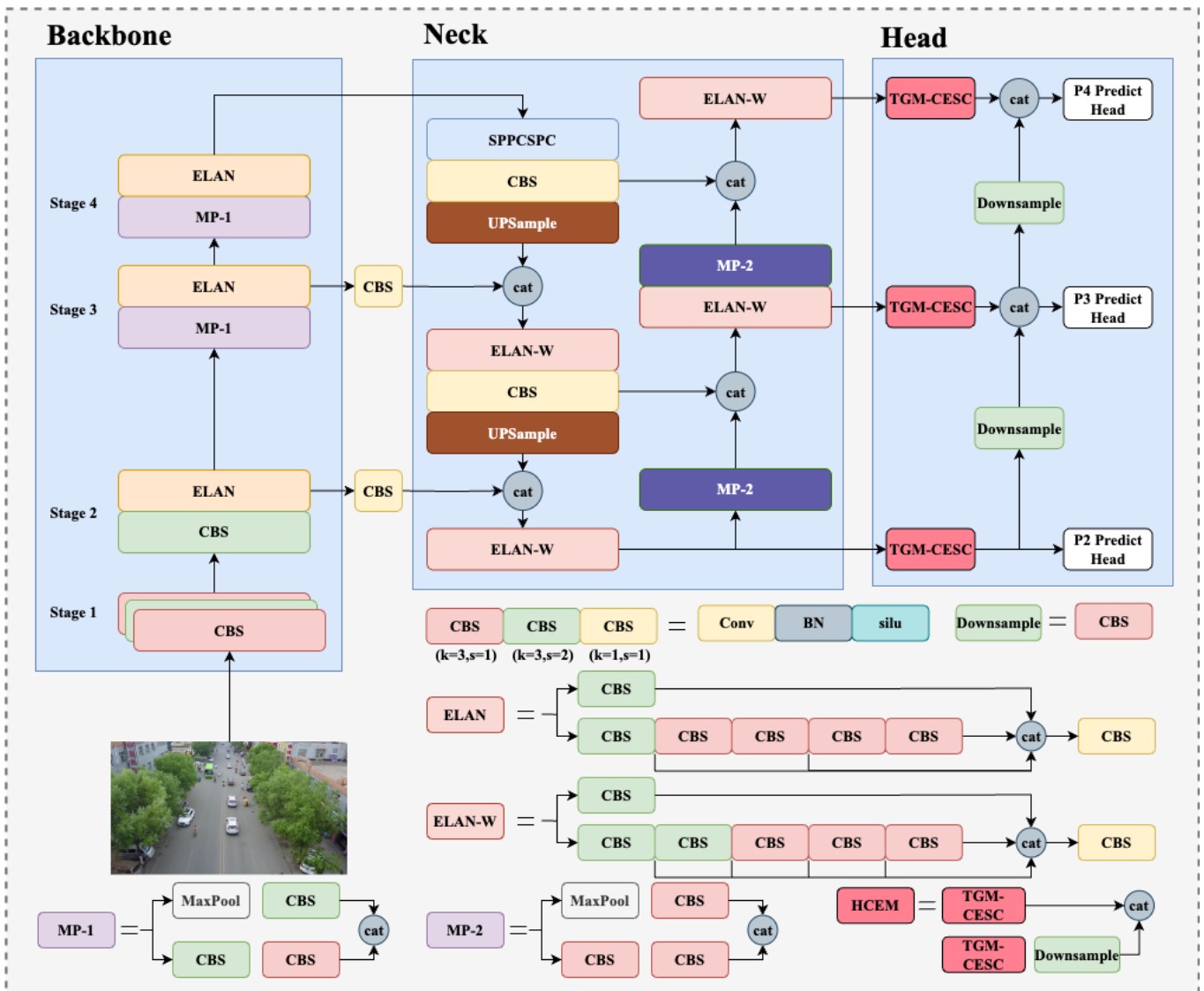

**Figure 2.** TheArchitecture of Efficient YOLOv7-Drone. In Efficient YOLOv7-Drone, we introduced the P2 detection head while discarding the less efficient P5 detection head. Additionally, the TGM-CESC module replaced the RepConv module, and the head context-enhanced method (HCEM) was incorporated in the detection head section.

### 3.1. Overview of YOLOv7

Consistently at the forefront of single-stage object detection, the YOLO series expertly balances detection accuracy with computational efficiency. In its most recent version, YOLOv7 introduces several innovative techniques, including re-parameterized convolutions, an efficient layer aggregation network, and a novel dynamic label assignment strategy. These advancements propel YOLOv7 beyond other detection algorithms in both accuracy and detection speed. Among the different variants of YOLOv7, such as YOLOv7-tiny and YOLOv7-X, we have chosen the standard YOLOv7 as our benchmark model, with an emphasis on its efficiency and real-time performance. However, its inherent design for general images suggests potential shortcomings when applied to aerial imagery. Acknowledging this, we have undertaken specific optimizations to the YOLOv7 architecture to better address the unique challenges posed by aerial imagery.

*3.2. Efficient YOLOv7-Drone*

Prior methods of object detection in aerial imagery predominantly adopted a coarse-to-fine strategy [17,19,20,39]. With transformers becoming increasingly popular in the realm of computer vision, recent studies [40–42] have sought to infuse self-attention mechanisms into the aerial image detection process. Although this integration elevates detection accuracy, the added stages and self-attention modules contribute to computational overhead, often making real-time detection elusive for many algorithms. Furthermore, the conventional practice of utilizing high-resolution images exacerbates this computational burden—a significant constraint for resource-limited drone platforms. In light of the challenges, we introduce the Efficient YOLOv7-Drone model, specifically optimized for low-resolution images. These are images resized to dimensions such as $640 \times 640$ pixels from their original high-resolution captured by drones. While such reduced pixel dimensions inherently offer fewer details compared to their original high-resolution state, they bring computational benefits crucial for real-time processing. With this approach, our model ensures real-time detection while still delivering outstanding detection accuracy.

**Introductionof the P2 Detection Head.** In drone aerial imagery, extremely small instances are prevalent. With low-resolution images as our input, the problem exacerbates as image scaling further reduces objects sizes. To mitigate this, we implemented the P2 detection head, designed explicitly for detecting minuscule objects. Figure 2 (P2 Prediction Head) illustrates this detection head sourcing feature maps from the Backbone's second stage. Distinct for its low-level and high-resolution qualities, this feature map retains more detailed information of tiny objects. Designed following conventions from other detection heads, this structure offers heightened sensitivity to tiny objects. Although this enhancement may pose greater computational requirements, the detection efficacy for tiny objects is markedly improved.

**P5Detection Head Omission and Network Refinements.** In aerial imagery with low-resolution inputs, there is often a prevalence of numerous small objects. The P5 detection head, which processes feature maps in the YOLO model, is subjected to a 32-fold down-sampling, emphasizing its high-level and low-resolution features. Such aggressive down-sampling considerably degrades object details, causing the P5 head to frequently misidentify small objects. This error in identifying small objects negatively impacts the overall accuracy of the detection. While the primary intent behind the P5 detection head was to detect larger objects, our observations reveal that eliminating it allows the P4 detection head to not only take over its responsibilities but also to enhance detection accuracy for smaller targets. In pursuit of greater model efficiency, we deemed Stage 5 of the Backbone superfluous and subsequently removed it. These modifications resulted in a 56.29% reduction in network parameters.

**CBSModule Channel Optimization.** As illustrated in Figure 2, the CBS module serves as an integral connector between the network's Backbone and Neck. The Backbone primarily extracts foundational features from the input images, while the Neck refines these features, enhancing their discriminatory power for object detection. By bridging these two components, CBS plays an essential role in reducing the dimensionality of feature maps and ensuring a smooth flow of information from the Backbone to the Neck. Specifically, the CBS module processes the feature maps extracted by Stage2 and Stage3 of the Backbone, and through a $1 \times 1$ convolution with a stride of 1, it reduces the channel count to a quarter of its original value, maintaining its spatial dimensions. Although this strategy improves computational efficiency, a significant reduction in channel count might lead to a loss of both detailed and high-level semantic information. To address this, we optimized the channel configuration of the CBS, ensuring that after processing, the channel count is only halved, preserving more information. In response to this adjustment in the CBS, discrepancies in channel count arose in the Neck section. Consequently, we made corresponding structural and channel count modifications to the Neck section, with details depicted in Figure 2 (Neck).

### 3.3. Context-Enhanced Sparse Convolution with Target-Guided Masking

#### 3.3.1. Overview of Sparse Convolution

Given the inherent hardware limitations of drone platforms, contemporary research endeavors to strike an optimal balance between accuracy and efficiency for aerial image object detection. Sparse convolution emerges as a viable solution to this predicament. This method emphasizes sparsely sampled regions via pixel-level masks, thereby mitigating the computational overhead associated with intricate backgrounds. In standard convolution operations, all pixels in a feature map receive uniform treatment, irrespective of their affiliation to either the foreground or background. Yet, in object detection, the focus predominantly rests on the target and its immediate vicinity. As depicted in Figure 1, the foreground usually constitutes a minor fraction of aerial images, implying that conventional convolution techniques engage in superfluous computations over expansive background regions. As underscored in [43], these background segments frequently harbor considerable noise, thereby undermining detection precision. Theoretically, sparse convolution can adeptly navigate these impediments. However, the success of sparse convolution greatly depends on the generated pixel-level masks. If these masks are inaccurately generated, the vital information contained within the omitted regions can significantly hamper the detection accuracy.

DynamicHead [44] leverages spatial gates to efficiently merge features across various scales. QueryDet [45] designed a cascade sparse query structure to reduce the model's complexity. However, most of these methods employ a fixed mask ratio to generate pixel-level masks. Given the significant variability in the pitch angle and altitude of aerial imagery, the proportion of the image foreground can vary significantly. Relying on a fixed mask ratio could lead to the loss of vital target information, consequently decreasing detection accuracy. Addressing this concern, CEASC [21] introduced the adaptive multi-layer masking (AMM) strategy. This strategy dynamically adjusts the mask ratio, striking a balance between detection precision and computational efficiency. However, the AMM strategy continues to rely on the mask ratio to dictate the generation of sparse convolution masks, using only this ratio as a constraint might introduce a high degree of randomness, potentially masking out vital information. To address this challenge, we introduce the TGM-CESC module in this study. Utilizing the target-guided mask methodology, we impose pixel-level constraints on the formation of sparse convolution masks.

#### 3.3.2. Target-Guided Mask in CESC Implementation

As described in [46], the surrounding background information of an instance plays a crucial role in object detection tasks. Therefore, [21] introduced the context enhancement (CE) module, which leverages both focal information and global context for feature enhancement, subsequently improving computational stability. Recognizing the efficiency of the CE module, we retained it along with its corresponding $L_{\text{norm}}$ loss function. Integrating this with our innovative target-guided mask strategy, we named the newly-formed module as TGM-CESC.

For the mask generation in sparse convolution, we followed [21] and employed the Gumbel-Softmax trick [47], formulated as follows:

$$\mathbf{P_i} = \begin{cases} \sigma(\text{Gumbel}(F_i)) > 0.5, & \text{For training} \\ F_i > 0, & \text{For inference} \end{cases} \tag{1}$$

Here, $\sigma$ denotes the sigmoid function, and $Gumbel(\cdot)$ refers to the addition of Gaussian noise to its inner value. The feature $F_i$ has the shape $\mathbb{R}^{B \times 1 \times H \times W}$ and is obtained through convolution with $W_{mask}$ of dimension $\mathbb{R}^{C \times 1 \times 3 \times 3}$. In accordance with Equation (1), only the sampling regions with a mask value of 1 are subjected to convolution during the inference phase, which results in decreased computational overhead and enhanced detection efficiency.

A critical aspect of sparse convolution lies in mask generation. Relying purely on ratios can yield unpredictable mask outcomes. To address this concern, we introduced the target-guided mask approach, as detailed in Algorithm 1. Specifically, after generating the masks, we use the label values from the corresponding original image to produce a pixel-wise binary map distinguishing foreground and background. This binary map corresponds pixel by pixel with the generated masks, allowing for more precise and efficient constraints on mask generation (as shown in Figure 3). We employ the following loss:

$$L_{\text{mask}}(P, T) = \frac{1}{L} \sum_{i=1}^{L} \left( 1 - \frac{2 \times \text{Intersection}(\mathbf{P_i}, \mathbf{T_i}) + \epsilon}{\text{Union}(\mathbf{P_i}, \mathbf{T_i}) + \epsilon} \right) \tag{2}$$

where $L$ denotes the amount of layers in Neck(FPN-PAN). $P_i$ represents the mask prediction for the $i^{th}$ layer, while $T_i$ corresponds to the ground truth for that mask. The functions Intersection$(\cdot)$ and Union$(\cdot)$ compute the intersection and union between the predicted mask and the ground truth, respectively.

As depicted in Figure 3, TGM-CESC operates as follows. Initially, the input feature value $F_i$ undergoes sparse convolution, yielding the feature value $S_i$ and a binary mask $P_i$. Using the label associated with the input feature, a binary mask $T_i$ is generated via the target-guided mask technique. Subsequently, $T_i$ with the $L_{\text{mask}}$ loss imposes pixel-level restrictions on the sparse convolution mask creation. To counteract the omission of background details around the target, $F_i$ is processed using PW convolution, securing its comprehensive information $G_i$. The mean and variance of the global feature $G_i$ facilitate the group normalization of $S_i$, intended to recover absent contextual data.

$$\mathbf{S}_i = w \times \frac{\mathbf{S}_i - \text{mean}[\mathbf{G}_i]}{\text{std}[\mathbf{G}_i]} + b \tag{3}$$

where mean[·] and std[·] denote the mean and standard deviation, respectively, and w and b are learnable parameters.

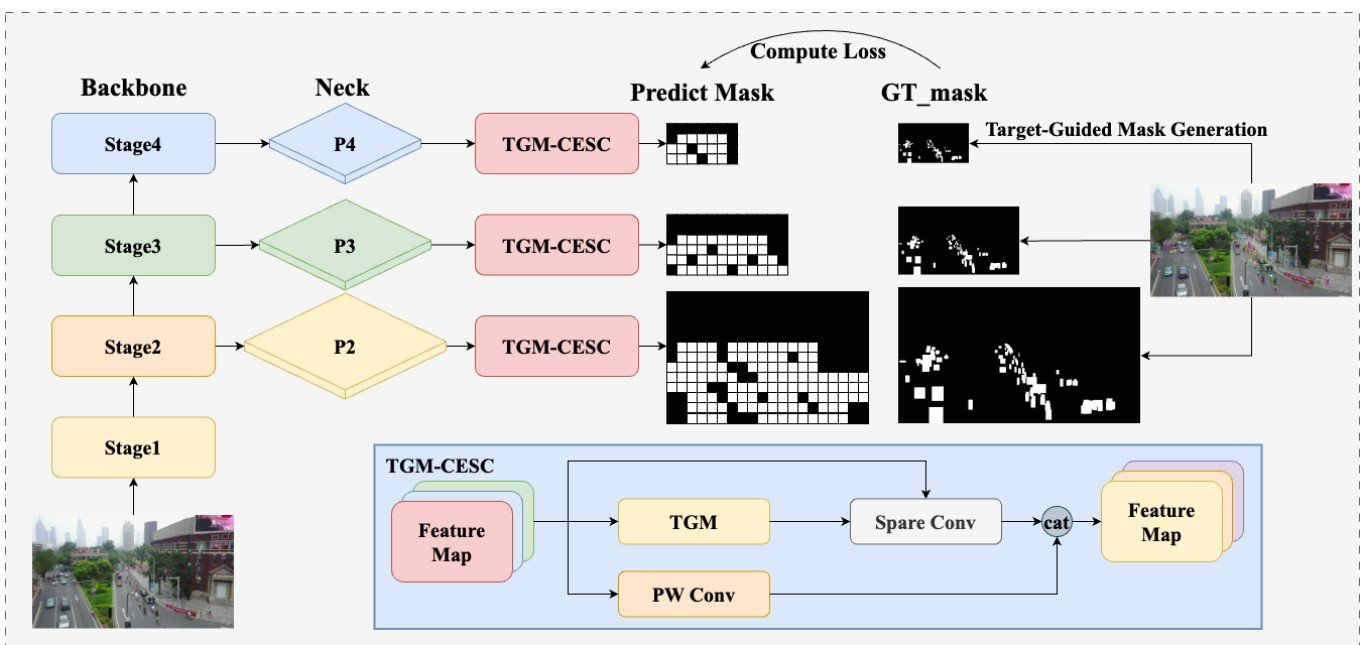

**Figure 3.** Operation of TGM-CESC in Efficient YOLOv7-Drone. TGM-CESC produces a distinct mask map for each layer in the Neck (FPN-PAN). Corresponding ground truth binary masks (GT_mask) are derived from object labels. By computing the loss between each mask map and its respective GT_mask, we achieve pixel-level constrained generation of sparse convolution masks.

---

**Algorithm 1** Target-Guided Mask Generation

---

**Require:** mask_map $M$, targets $T$
**Ensure:** GT_mask
 1: GT_mask ← []
 2: **for** each feature layer $L$ in $M$ **do**
 3:     binary_map_layer ← new zero tensor with shape of $L$
 4:     **for** each image $I$ in batch of $L$ **do**
 5:         this_target ← get targets for current image from $T$
 6:         **if** this_target is empty **then**
 7:             continue
 8:         **end if**
 9:         txywh ← scale target coordinates using dimensions of $I$
10:         txyxy ← convert txywh to top-left and bottom-right coordinates
11:         binary_map ← new zero tensor with shape of $I$
12:         **for** each box $B$ in txyxy **do**
13:             convert $B$ to integer coordinates
14:             set pixels in binary_map inside $B$ to 1
15:         **end for**
16:         append binary_map to binary_map_layer
17:     **end for**
18:     append binary_map_layer to GT_mask
19: **end for**
20: **return** GT_mask

---

To diminish the data loss from sparse convolution and stabilize training, we employ both sparse and traditional dense convolutions during training. The feature map is derived as $C_i$ when input through dense convolution. Thereafter, $C_i$ augments the sparse feature map $S_i$ by optimizing the MSE loss.

$$\mathcal{L}_{\text{norm}} = \frac{1}{L} \sum_{i=1}^{L} \|\mathbf{C}_i \times \mathbf{P}_i - \mathbf{S}_i\|^2 \tag{4}$$

where $L$ denotes the number of layers in Neck(FPN-PAN).

By minimizing $L_{\text{mask}}$, we can impose pixel-wise constraints on the generation of sparse convolution masks, thereby facilitating the production of more accurate and efficient masks. When combined with the traditional $L_{\text{det}}$ loss and the $L_{\text{norm}}$ loss from CE, the final training loss function is given by:

$$L = L_{\text{det}} + L_{\text{norm}} + L_{\text{mask}} \tag{5}$$

To ensure the focus of our model primarily on the foreground of drone-captured images and to mitigate the influence of background noise on detection precision. As shown in Figure 2, we have integrated the TGM-CESC module into the detection head of the Efficient YOLOv7-Drone. Taking into consideration the balance between detection accuracy and computational efficiency, we adopted the TGM-CESC in place of the original re-parameterized convolution (RepConv) module.

### 3.4. Head Context-Enhanced Method (HCEM)

The success of sparse convolution detection is intrinsically linked to the quality of mask generation. While the TGM-CESC method allows us to impose constraints on the mask at the pixel level, the intricate backgrounds inherent in aerial imagery may result in the omission of crucial data. To mitigate this, we introduced the head context-enhanced method (HCEM), as illustrated in Figure 2. This module, by leveraging feature maps from TGM-CESC across adjacent layers, strives to glean more comprehensive semantic insights. Therefore, it alleviates the information loss in low-resolution detection layers caused by mask generation quality. More specifically, we extract feature maps (P2, P3, P4) from different layers of the FPN using the TGM-CESC module. Recognizing the

detailed richness inherent in low-level, high-resolution feature maps, we deploy an upward propagation mechanism: P2 is first concatenated (Cat($\cdot$)) with P3, followed by the upward propagation of the resultant feature map for further concatenation with P4. Before merging the adjacent features, we downsampled the feature maps to ensure they have the same scale. The operation can be represented by the following formula:

$$\mathbf{T_i} = \mathrm{Cat}(\mathrm{Downsample}(\mathbf{P_{i-1}}), \mathbf{P_i}) \qquad (6)$$

where Downsample($\cdot$) employs a CBS($\cdot$) module with a stride of 2, while $P_i$ represents the feature map from the $i^{th}$ layer obtained through the TGM-CESC module. Cat($\cdot$) denotes the channel-wise concatenation operation. To avoid additional computational overhead within HCEM, we exclusively use one CBS module for downsampling and one cat operation for feature fusion. No other extra computations have been introduced.

## 4. Experiments

### 4.1. Datasets and Evaluation Measures

To demonstrate the effectiveness of our proposed method, we conducted extensive experiments on two primary drone aerial object detection benchmark datasets, namely Visdrone [48] and UAVDT [49]. These datasets were selected due to their diverse data representation, encompassing aerial images from various weather conditions, terrains, and objects spanning different traffic and daily life scenarios. They provide meticulous annotations for each image and feature challenging scenarios, including object occlusions, small targets, target overlaps, and intricate backgrounds. Furthermore, their widespread use in the aerial image detection domain ensures a credible and pertinent evaluation of our method.

**VisDrone dataset.** This dataset comprises 288 video clips and 10,209 high-resolution static images. Of these, 6471 are allocated for training, 548 for validation, and 3190 for testing. The image resolutions range from $960 \times 540$ to $2000 \times 1500$. Captured across 14 different cities, the dataset encompasses a myriad of shooting scenarios, covering 10 distinct target categories, namely pedestrian, people, bicycle, car, van, truck, tricycle, awning-tricycle, bus, and motor. Due to its pronounced class and size imbalances, it serves as an ideal benchmark for studying small object detection challenges. To ensure consistency with prior research, all test results are based on the validation set.

**UAVDT dataset.** Compared to the VisDrone dataset, UAVDT offers an even more extensive collection of drone-captured imagery. It contains 23,258 images for training and 15,069 images for testing, all with a resolution of $1024 \times 540$. This dataset focuses on three categories: buses, trucks, and cars.

**Evaluation measures.** In our study, we adopted mean average precision (mAP), average precision (AP), and average recall (AR) for accuracy evaluation. For efficiency, we considered GFLOPs, FPS, and the total parameter count.

### 4.2. Implementation Details

We implemented our model using PyTorch. All of our experiments were conducted on a single NVIDIA RTX 3090 GPU for both training and testing. During the training phase, we leveraged partial weights from a pre-trained YOLOv7 model, considerably reducing the training time. To ensure consistency and fairness in our experiments, we trained our model for 200 epochs and fixed the batch size at 8. For efficiency, we standardized the input width and height to 640. We utilized the SGD optimizer, with all other parameters set to the default configurations of YOLOv7.

### 4.3. Comparison with the State of the Art

We conducted experiments on the VisDrone and UAVDT datasets, comparing our method to state-of-the-art object detectors. To emphasize our method's detection performance efficiency, we refrained from using additional tricks during inference. Our model employs a backbone named "Modified ELANNet". Table 1 shows that the Modified

ELANNet has only 51% of ResNet's parameters, and its GFLOPs are also lower than those of ResNet50. Most of the comparative methods rely on high-resolution images and are multi-stage detectors. To maintain fairness in the experiment, the backbone in the selected methods was of similar complexity or even more intricate than ours.

On the VisDrone dataset, we compared our method against ten recent popular methods. Specifically, RetinaNet [7], ClusDet [20], DMNet [19], GLSAN [50], QueryDet [45], CascadeNet [51], and CascadeNet+MF [51] utilize ResNet-50 as their Backbone. GFL V1 [37], which incorporates the CEASC [21] structure, employs ResNet-18, while HRD-Net [39] uses both ResNet-18 and ResNet-101. DFPN [52] chose Modified CSP v5-M as its Backbone. As shown in Table 2, even though our approach uses a lower resolution image as input, it achieved the best results across all three main evaluation metrics. This outcome convincingly demonstrates our technique's ability to balance detection accuracy with enhanced efficiency.

To underscore the outstanding performance and resilience of our model, we configured its input to a resolution of 1280 × 1280. Table 3 illustrates that our model attains scores of 40.9%, 63.5%, and 43.6% across the three key evaluation metrics. This performance substantially surpasses that of other state-of-the-art methods, even when they utilize more complex Backbones and superior resolutions.

For the UAVDT dataset, we benchmarked our method against ClusDet [20], DM-Net [19], DFPN [52], and ARMNet [53]. As demonstrated in Table 4, even when working with a lower image resolution, our approach excels across all three primary evaluation metrics. This reaffirms our method's prowess in seamlessly integrating detection accuracy with heightened efficiency.

**Table 1.** Comparative Analysis of Parameters and GFLOPs Among Various Backbone Architectures.

| BackBone | Param | GLOPS |
|---|---|---|
| ResNet18 | 11.18M | **29.78** |
| ResNet50 | 23.50M | 67.45 |
| ResNet101 | 42.50M | 128.39 |
| ResNext101_32× 4d | 42.13M | 131.48 |
| ResNext101_64× 4d | 81.41M | 254.42 |
| Modified ELANNet | **6.02M** | 59.53 |

**Table 2.** Comparison of Efficient YOLOv7-Drone against other state-of-the-art methods using a small Backbone on the VisDrone validation set.

| Method | BackBone | Resolution | AP[%] | AP50[%] | AP75[%] |
|---|---|---|---|---|---|
| RetinaNet [7] | ResNet-50 | 2400 × 2400 | 26.2 | 44.9 | 27.1 |
| ClusDet [20] | ResNet-50 | 1000 × 600 | 26.7 | 50.6 | 24.4 |
| DMNet [19] | ResNet-50 | 1000 × 600 | 28.2 | 47.6 | 28.9 |
| GLSAN [50] | ResNet-50 | 1000 × 600 | 25.8 | 51.5 | 22.9 |
| HRDNet [39] | ResNet-18+ResNet-101 | 2666 × 1600 | 28.3 | 49.3 | 28.2 |
| QueryDet [45] | ResNet-50 | 2400 × 2400 | 28.3 | 48.1 | 28.8 |
| GFL V1 (CEASC) [21] | ResNet-18 | 1333 × 800 | 28.7 | 50.7 | 28.4 |
| CascadeNet [51] | ResNet-50 | - | 28.8 | 47.1 | 29.3 |
| DFPN [52] | Modified CSP v5-M | 768 × 768 | 30.3 | 51.9 | 30.5 |
| AMRNet [53] | ResNeXt-50 | 1500 × 800 | 31.7 | 52.6 | **33.1** |
| Efficient YOLOv7-Drone(Ours) | Modified ELANNet | **640 × 640** | **32.1** | **53.6** | 32.5 |

**Table 3.** Comparison of Efficient YOLOv7-Drone against other state-of-the-art methods using a large backbone on the VisDrone validation set. 'MF' stands for model fusion. The ★ denotes the multi-scale inference.

| Method | BackBone | Resolution | AP[%] | AP50[%] | AP75[%] |
|---|---|---|---|---|---|
| ClusDet [20] | ResNeXt-101 | $1000 \times 600$ | 28.4 | 53.2 | 26.4 |
| ClusDet★ [20] | ResNeXt-101 | $1000 \times 600$ | 32.4 | 56.2 | 31.6 |
| DMNet [19] | ResNeXt-101 | $1000 \times 600$ | 29.4 | 49.3 | 30.6 |
| HRDNet★ [39] | ResNeXt-50+ResNeXt-101 | $3800 \times 2800$ | 35.5 | 62.0 | 35.1 |
| CascadeNet+MF★ [51] | ResNet-50 | - | 30.1 | 58.0 | 27.5 |
| AMRNet★ [53] | ResNeXt-101 | $1500 \times 800$ | 36.7 | - | - |
| Efficient YOLOv7-Drone(Ours) | Modified ELANNet | **1280 × 1280** | 39.9 | 62.5 | 42.1 |
| Efficient YOLOv7-Drone★(Ours) | Modified ELANNet | **1280 × 1280** | **40.9** | **63.5** | **43.6** |

**Table 4.** Comparison results of Efficient YOLOv7-Drone with other state-of-the-art methods on the UAVDT dataset.

| Method | BackBone | Resolution | AP[%] | AP50[%] | AP75[%] |
|---|---|---|---|---|---|
| ClusDet [20] | ResNet-50 | $1000 \times 600$ | 13.7 | 26.5 | 12.5 |
| DMNet [19] | ResNet-50 | $1000 \times 600$ | 14.7 | 24.6 | 16.3 |
| GLSAN [50] | ResNet-50 | $1000 \times 600$ | 17.0 | 28.1 | 18.8 |
| DFPN [52] | Modified CSP v5-M | $640 \times 640$ | 17.1 | 29.3 | 18.1 |
| ARMNet [53] | ResNet-50 | $1500 \times 800$ | 18.2 | 30.4 | 19.8 |
| Efficient YOLOv7-Drone(Ours) | Modified ELANNet | **640× 640** | **20.0** | **36.0** | **20.1** |

### 4.4. Ablation Study

To further validate the effectiveness of our proposed model, we conducted an extensive ablation study on the VisDrone dataset. For brevity and clarity, we labeled our modifications as: introduction of the P2 detection head as 'A', removal of the P5 detection head as 'B', CBS module channel optimization as 'C', target-guided context enhancement sparse convolution (TGM-CESC) as 'D', and head context-enhanced method (HCEM) as 'E'.

#### 4.4.1. Comparison with the Baseline Model

We carried out a comprehensive evaluation on the VisDrone validation set to accurately assess the performance of each introduced component. Our evaluation took into account both detection accuracy and efficiency, utilizing a diverse set of metrics: mAP, AP50, AP75, GPU memory consumption, giga floating-point operations (GFLOPs), and FPS.

As detailed in Table 5 and compared to the baseline model:

- The introduction of the P2 detection head (A) led to improved detection metrics. Specifically, mAP, AP50, and AP75 increased by 0.7%, 0.8%, and 1.2%, respectively. However, this also introduced additional memory and computation overhead.
- By removing the P5 detection head (B), we noted improvements in mAP, AP50, AP75, and AR, which reached 30.0%, 51.5%, 30.1%, and 52.8%, respectively. Additionally, this change reduced the GPU memory overhead by 56.32% and decreased GFLOPs from 118.2 to 115.2, resulting in an FPS boost of 29.68%.
- The CBS module channel optimization (C) slightly increased memory and GFLOPs but boosted the mAP, AP50, and AP75 to 31.0%, 52.2%, and 31.6%, respectively.
- With the integration of the TGM-CESC module (D), the model focused on image foreground areas, achieving mAP, AP50, and AP75 scores of 31.5%, 52.8%, and 32.1%. By utilizing the sparse convolution, the GFLOPs was reduced to 133.0. However, substituting the well-performing RepConv module during the inference phase led to a slight decrease in FPS.

**Table 5.** Ablation study results on the VisDrone dataset. The components are labeled as: A—Introduction of the P2 detection head, B—Removal of the P5 detection head, C—CBS module channel optimization, D—Target-guided context enhancement sparse convolution (TGM-CESC), and E—Head context-enhanced method (HCEM).

| A | B | C | D | E | mAP[%] | AP50[%] | AP75[%] | AR[%] | Param | GFLOPs | FPS (Frames/sec) |
|---|---|---|---|---|--------|---------|---------|-------|-------|--------|------------------|
| | | | | | 28.6 | 49.5 | 28.4 | 48.9 | 36.53M | **103.3** | **212.76** |
| ✓ | | | | | 29.3 | 50.3 | 29.6 | 52.0 | 40.46M | 118.2 | 136.99 |
| ✓ | ✓ | | | | 30.0 | 51.5 | 30.1 | **52.8** | **17.68M** | 115.2 | 166.67 |
| ✓ | ✓ | ✓ | | | 31.0 | 52.2 | 31.6 | 51.9 | 19.16M | 153.1 | 138.89 |
| ✓ | ✓ | ✓ | ✓ | | 31.5 | 52.8 | 32.1 | 51.9 | 29.99M | 133.0 | 96.15 |
| ✓ | ✓ | ✓ | ✓ | ✓ | **32.1** | **53.6** | **32.5** | 52.2 | 31.79M | 144.6 | 89.29 |

Finally, with the introduction of HCEM (E), we achieved significant performance improvements while only incurring a slight increase in memory and computational overhead. Compared to the baseline, there was an uplift in mAP, AP50, AP75, and AR by 3.5%, 4.1%, 4.1%, and 3.3%, respectively.

For performance improvements of less than 1%, it is essential to ensure the reliability and consistency of these incremental gains. To rigorously ascertain the validity of these marginal enhancements and to mitigate potential overfitting or random variations, we utilized $k$-fold cross-validation with $k = 5$. This method offers a comprehensive assessment of the model's resilience across diverse data subsets. For this validation, we combined the VisDrone training and validation datasets, resulting in a total of 7019 images, with 5615 images from the training set and 1404 from the validation set. Specifically, we applied $k$-fold cross-validation to models incorporating the TGM-CESC and HCEM modules. For comparative analysis, we similarly conducted $k$-fold cross-validation on models without these modules. Tables 6–8 display the results, which indicate that the TGM-CESC and HCEM modules substantially improved the detection accuracy of the model.

**Table 6.** Detection accuracy results of the model without integrating the TGM-CESC and HCEM modules, using $k$-fold cross-validation on the VisDrone Dataset.

| Validation Fold | mAP[%] | AP50[%] | AP75[%] |
|-----------------|--------|---------|---------|
| 1-Fold | 33.2 | 56.6 | 33.7 |
| 2-Fold | 32.2 | 55.2 | 32.4 |
| 3-Fold | 32.5 | 55.9 | 32.7 |
| 4-Fold | 33.0 | 56.4 | 33.1 |
| 5-Fold | 32.3 | 55.3 | 32.8 |
| Average | 32.6 | 55.9 | 33.0 |

**Table 7.** Detection accuracy results of the model with the integrated TGM-CESC module using $k$-fold cross-validation on the VisDrone Dataset.

| Validation Fold | mAP[%] | AP50[%] | AP75[%] |
|-----------------|--------|---------|---------|
| 1-Fold | 33.6 | 56.9 | 34.3 |
| 2-Fold | 32.7 | 55.6 | 33.3 |
| 3-Fold | 33.0 | 56.6 | 33.2 |
| 4-Fold | 33.4 | 56.7 | 33.8 |
| 5-Fold | 33.1 | 56.2 | 33.5 |
| Average | 33.1 | 56.4 | 33.6 |

**Table 8.** Detection accuracy results of the model with the integrated HCEM module using *k*-fold cross-validation on the VisDrone Dataset.

| Validation Fold | mAP[%] | AP50[%] | AP75[%] |
|:---:|:---:|:---:|:---:|
| 1-Fold | 33.8 | 57.3 | 34.6 |
| 2-Fold | 32.6 | 55.8 | 32.9 |
| 3-Fold | 33.1 | 56.6 | 33.3 |
| 4-Fold | 33.6 | 57.1 | 34.2 |
| 5-Fold | 33.3 | 56.6 | 33.9 |
| Average | 33.3 | 56.7 | 33.8 |

4.4.2. Visualization between Baseline and the Efficient YOLOv7-Drone

From the experimental data, it is clear that the Efficient YOLOv7-Drone surpasses the baseline on the VisDrone dataset. To provide a more vivid and direct comparison between these two models, we have visualized their prediction results on the VisDrone dataset in Figure 4. The first column displays the ground truths of the images, the second offers predictions from the baseline, and the third column highlights the predictions from the Efficient YOLOv7-Drone. To emphasize the distinctions, we have magnified the regions with significant prediction differences. Observing the first rows of the images, it is evident that our approach has made significant improvements in reducing false detections compared to the baseline. In the second row, when dealing with scenes filled with small, densely packed objects, our model clearly outperforms, showcasing superior detection accuracy.

4.4.3. Details of the Efficient YOLOv7-Drone Design

**Introduction of the P2 Detection Head**. Aerial imagery often contains numerous small objects. When using low-resolution images as input, these objects can become extremely minute, commonly referred to as "tiny objects", which pose detection challenges. To address this and enhance detection of tiny objects, we incorporated features from the Stage 2 output of the Backbone, which leverages its rich and comprehensive representation of small objects. As demonstrated in Table 9, the introduction of the P2 detection head resulted in a 0.9% improvement in detection accuracy for small objects, highlighting its crucial role in our model.

**Table 9.** Comparison of Detection Accuracy between Baseline and the Introduction of the P2 Detection Head (A).

| Method | mAP[%] | AP50[%] | AP75[%] | $mAP_s$[%] | $mAP_m$[%] | $mAP_l$[%] |
|:---:|:---:|:---:|:---:|:---:|:---:|:---:|
| Baseline | 28.6 | 49.5 | 28.4 | 19.4 | 40.3 | 59.2 |
| +A | 29.3 | 50.3 | 29.6 | 20.3 | 40.5 | 58.0 |

**P5 Detection Head Omission: Enhancing Efficiency and Accuracy**. As shown in Table 10, following the introduction of the P2 detection head, we evaluated specific metrics for various detection heads. Notably, the P2 detection head exclusively focuses on detecting small objects, which aligns seamlessly with our initial purpose for its inclusion. Compared to the P4 detection head, the P5 detection head shows only a slight advantage specifically in the detection of larger objects. This difference arises from the initial allocation of anchors. In essence, if the P5 detection head were to be removed, the P4 detection head would be poised to assume its role. Consequently, to balance accuracy and computational efficiency, we opted to exclude the P5 detection head. As demonstrated in Table 5, this exclusion led to improvements in our model's mAP, AP50, and AP75 metrics by 0.7%, 1.2%, and 0.5%, respectively. Concurrently, the model achieved a significant 56.29% reduction in the number of parameters, enhancing computational efficiency and improving FPS performance.

**Table 10.** Metrics for Various Detection Heads After Introducing the P2 Detection Head.

| Method | $mAP_s$[%] | $mAP_m$[%] | $mAP_l$[%] |
|---|---|---|---|
| P5 Detection Head | 7.2 | 32.5 | 58.2 |
| P4 Detection Head | 10.9 | 35.3 | 42.3 |
| P3 Detection Head | 15.9 | 21.9 | 0.8 |
| P2 Detection Head | 12.2 | 8.3 | 0.0 |

Table 11 presents a comparison of various metrics for the P4 detection head before and after the removal of the P5 detection head. As the table illustrates, after removing the P5 detection head, all metrics associated with the P4 detection head display notable improvements. Specifically, the $mAP_l$ value sees an increase of 10.2%. Although this metric remains marginally lower than that of the P5 detection head before its removal, the enhancement in other metrics sufficiently compensates for this difference. These experimental results confirm that, without the P5 detection head, the P4 detection head can effectively assume its responsibilities and even surpass its performance.

**Table 11.** Comparison Between P4 Detection Head and After P5 Removal on the VisDrone dataset.

| Method | mAP[%] | AP50[%] | AP75[%] | $mAP_s$[%] | $mAP_m$[%] | $mAP_l$[%] |
|---|---|---|---|---|---|---|
| P4 Detection Head | 20.9 | 32.2 | 22.8 | 10.9 | 35.3 | 42.3 |
| After P5 Removal (P4 Detection Head) | 24.7 | 38.4 | 26.4 | 13.8 | 38.7 | 52.5 |

**CBS Module Channel Optimization**: To enhance the transfer efficiency of features from the Backbone to the Neck, we employed the CBS module channel optimization method. As shown in Table 12, this optimization led to increments of 1.0% in mAP, 0.7% in AP50, 1.5% in AP75, 0.9% in $mAP_s$, 1.0% in $mAP_m$, and a significant 1.9% in $mAP_l$. The empirical results underscore the effectiveness of the CBS channel optimization in better preserving feature information.

**Table 12.** Ablation on the Impact of CBS Module Channel Optimization. The components are labeled as: A—Introduction of the P2 detection head, B—Removal of the P5 detection head, C—CBS module channel optimization.

| Method | mAP[%] | AP50[%] | AP75[%] | $mAP_s$[%] | $mAP_m$[%] | $mAP_l$[%] |
|---|---|---|---|---|---|---|
| Baseline + A + B | 30.0 | 51.5 | 30.1 | 21.1 | 41.2 | 52.2 |
| Baseline + A + B + C | 31.0 | 52.2 | 31.6 | 22.0 | 42.2 | 54.1 |

### 4.4.4. Details of the TGM-CESC Analysis

To ensure our model emphasizes the image's foreground, we integrated the TGM-CESC module. Table 13 presents results from two configurations: one adding TGM-CESC after RepConv and another replacing RepConv entirely with TGM-CESC. The latter configuration showed slightly better accuracy but experienced a 1% decrease in the AR score. Nonetheless, considering overall detection efficiency, we chose to substitute RepConv with TGM-CESC.

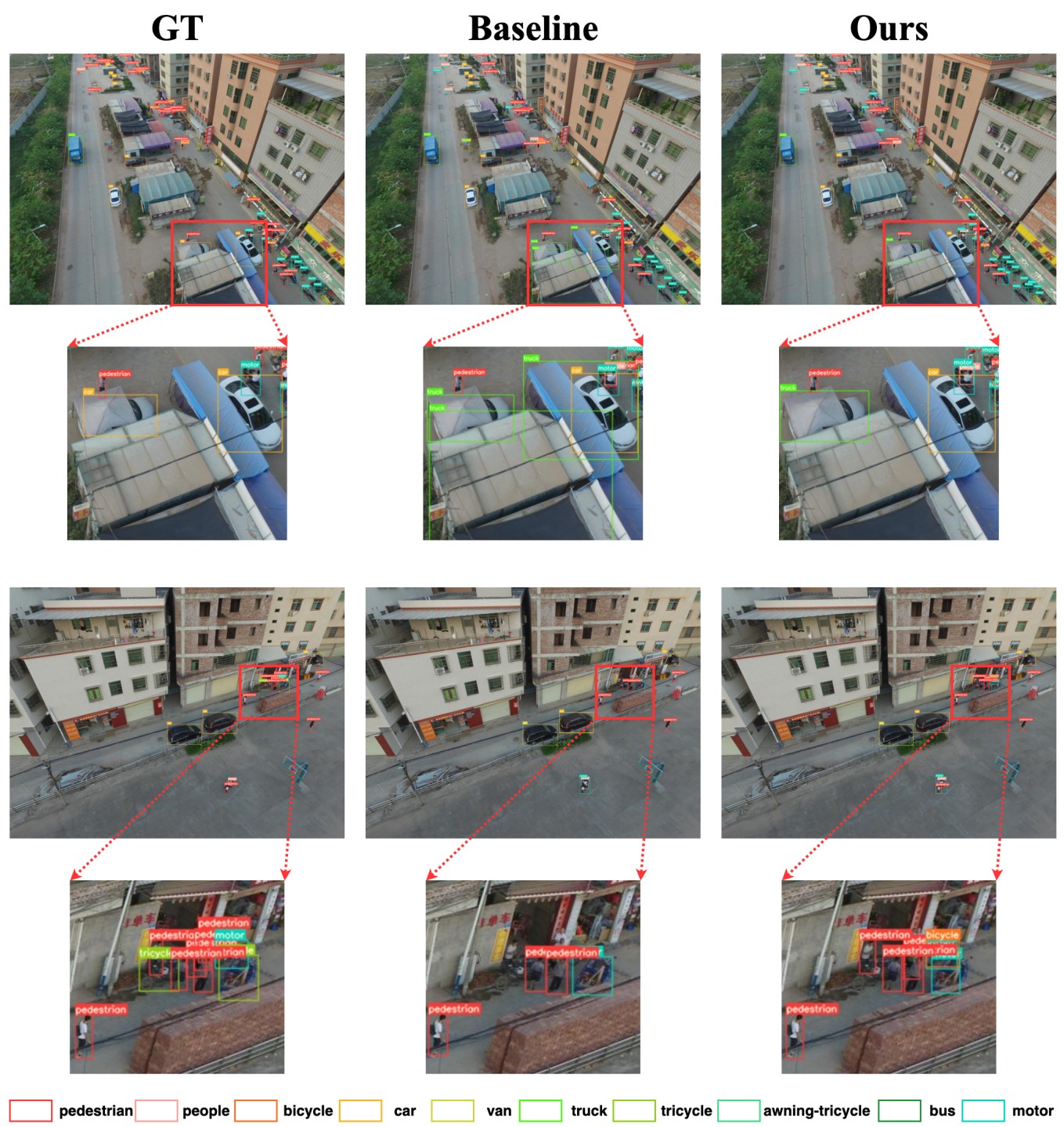

**Figure 4.** Comparative analysis of baseline vs. Efficient YOLOv7-Drone(Ours) on the VisDrone Dataset.

**Table 13.** Ablation study on the placement of the TGM-CESC module.

| Method | mAP[%] | AP50[%] | AP75[%] | AR[%] |
|---|---|---|---|---|
| RepConv + TGM | 29.2 | 50.4 | 29.0 | 51.4 |
| TGM Only | 29.4 | 50.6 | 29.4 | 50.4 |

To underscore the significance of pixel-level constraints in TGM for sparse convolution mask generation, we evaluated its efficacy against the AMM module and another method with a mask ratio set to zero. As Table 14 indicates, the TGM method surpasses the other two techniques. Notably, in the AR metric, TGM achieves a 3.5% improvement over AMM. This improvement results from the pixel-level constraint of TGM, which promotes accurate mask generation while minimizing the chance of masking crucial details. Both AMM and TGM markedly outperform the fixed mask ratio approach, emphasizing the significance of focusing on the image foreground.

**Table 14.** Comparative Analysis on Constraint Methods for Sparse Convolutional Mask Generation.

| Method | mAP[%] | AP50[%] | AP75[%] | AR[%] |
| --- | --- | --- | --- | --- |
| Mask Ratio = 0 | 23.3 | 41.1 | 22.8 | 41.9 |
| AMM | 28.0 | 47.4 | 28.0 | 46.9 |
| TGM | 29.4 | 50.6 | 29.4 | 50.4 |

To clearly demonstrate our method's superiority, we visualized the three methods. As depicted in Figure 5, across different feature layers, the TGM method effectively directs the model's focus toward the image's foreground, outperforming the other two methods.

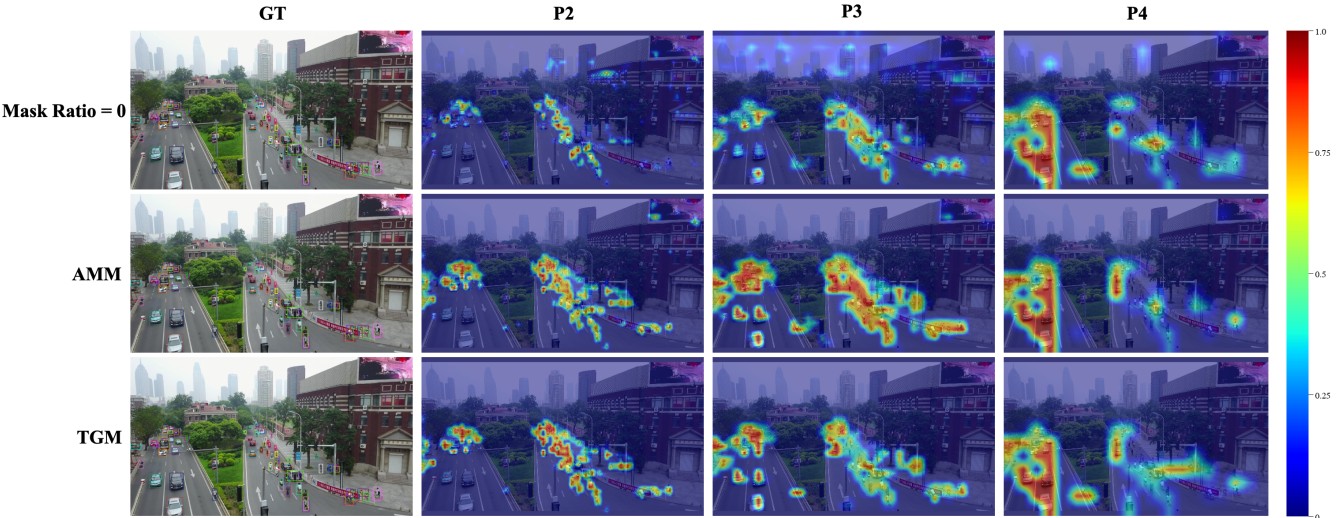

**Figure 5. Heatmaps of Attention Distribution across Layers 'P2' to 'P4'.** Row 1: Mask Ratio=0, Row 2: AMM, Row 3: TGM.

### 4.4.5. Details of the HCEM

Table 2 reveals that the introduction of the head context-enhanced method (HCEM) resulted in noticeable improvements in mAP, AP50, AP75, and AR metrics, increasing by 0.6%, 0.8%, 0.4%, and 0.3%, respectively. To offer a deeper insight into the HCEM's impact on our model, we visualized its effects. As shown in Figure 6, the feature information at each detection layer was substantially enriched with the integration of HCEM. This underscores HCEM's potential in counteracting the detail loss that comes with the use of sparse convolution.

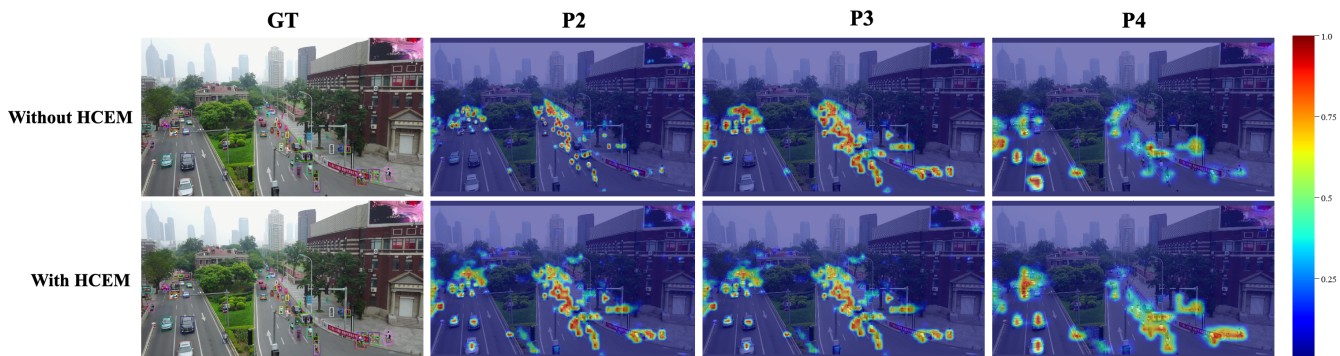

**Figure 6.** **Heatmaps of Attention Distribution across Layers 'P2' to 'P4'.** Row 1: Without HCEM, Row 2: With HCEM.

## 5. Discussion and Conclusions

In this study, we address three of the critical challenges in aerial image object detection. Targeting platforms powered by small drones with constrained computational capabilities, we present the Efficient Yolov7-Drone—an object detection algorithm boasting high precision and real-time performance. Unlike previous studies that primarily relied on computationally intensive high-resolution images, our approach capitalizes on low-resolution inputs. We surmise that if an algorithm performs effectively on low-resolution images, it will undoubtedly excel on high-resolution ones. As a result, our proposed algorithm is marked by its low-resolution, low-latency, and high precision and is tailored for drone platforms.

In our analysis, we observed that the process of downscaling high-resolution images often reduces small objects to "tiny" objects. This means that aerial images, which originally have a large number of small targets, now possess an abundance of these tiny objects. To address this, we integrated the P2 detection head, leveraging the detailed information from low-level, high-resolution feature maps to enhance the detection accuracy of these tiny objects. Notably, the P2 detection head introduces computational overhead. Furthermore, high-level, low-resolution images tend to lose crucial details of tiny objects, a phenomenon that is particularly pronounced in aerial images. This omission can result in erroneous object detection, affecting the overall accuracy. Such insights rendered the P5 detection head superfluous, prompting its removal along with the associated Stage 5 from the Backbone. Subsequently, we discerned that the CBS module, which bridges the Backbone and Neck, decreases the feature channel count from the Backbone by a factor of four. To safeguard against the potential loss of intricate details and semantic richness, we implemented a CBS channel optimization technique. Following this optimization, the channel count reduced by half. Additionally, we refined the architecture of the Neck segment to ensure proper channel alignment.

Given the notably low foreground proportion in drone aerial imagery and the potential detrimental effects of excessive background details on detection precision, we integrated the TGM-CESC module. Utilizing the advantages of sparse convolution, this module directs the model's attention predominantly towards the image's foreground. As part of this approach, the TGM method was developed to refine the generation of sparse convolution masks.

To address the potential masking of foreground information by sparse convolution, we devised the HCEM module. This module combines detailed, low-level, high-resolution feature maps with high-level, low-resolution maps, enhancing semantic comprehension. This integration helps restore any foreground details potentially masked by sparse convolution.

We validated the efficacy of our methodology through rigorous experiments using two popular drone aerial object detection benchmarks: VisDrone and UAVDT. Compared to other state-of-the-art methods evaluated on these datasets, our approach demonstrated superior performance.

Our approach is distinguished by its attention to the low foreground proportion in aerial images and its proficiency in balancing detection precision with efficiency. Notably, even with lower-resolution images, our method consistently delivers superior detection results. Additionally, while our enhancements were developed with YOLOv7 in mind, their applications are broader. Detectors addressing dense environments and numerous small targets could benefit from integrating our P2 detection head, foregoing the P5 detection head, and adopting the CBS channel optimization to align detection precision with efficiency. Likewise, detectors employing sparse convolution with an emphasis on foreground focus might find our TGM-CESC module advantageous.

**Author Contributions:** Conceptualization, X.F. and G.W.; methodology, X.F. and G.W.; software, G.W.; validation, X.F.; formal analysis, X.F., G.W. and X.Y.; investigation, X.F., G.W., X.Y., Y.L. and Y.B.; resources, X.Y. and Y.L.; data curation, X.F. and G.W.; writing—original draft preparation, X.F. and G.W.; writing—review and editing, X.Y., Y.L. and Y.B.; funding acquisition, Y.L. All authors have read and agreed to the published version of the manuscript.

**Funding:** This research was funded by the National Natural Science Foundation of China, grant number 12071218.

**Data Availability Statement:** The datasets analyzed in this study are publicly available. Specifically, the VisDrone and UAVDT datasets can be accessed through their respective official channels. Additionally, for convenience, we have provided download links for these datasets on our GitHub repository. The repository can be accessed at: https://github.com/GoodGoodStudyGT/Efficient_YOLOv7_Drone (accessed on 30 August 2023).

**Conflicts of Interest:** The authors declare no conflict of interest.

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
