# Peer review of "Efficient YOLOv7-Drone: An Enhanced Object Detection Approach for Drone Aerial Imagery"

_drones, doi:10.3390/drones7100616_

Round 1

Reviewer 1 Report

Summary:

In the reviewed article, the authors analyzed the problem of object detection in images acquired with unmanned aerial vehicles using algorithms using deep learning techniques. The authors of the publication note that the specifics of the problem of object detection in drone camera images, is significantly different from typical solutions presented in the literature. Input images contain extensive background areas, while objects are often small and often concentrated in dense groups. The authors proposed a solution to achieve a balance between detection efficiency and computational efficiency. The proposed solution is based on the YOLOv7 architecture but makes significant changes to its structure.

The proposed solution (YOLOv7-Drone) was compared with other state-of-the-art DL algorithms using two popular datasets - VisDrone and UAVDT.

The authors showed that the proposed solution not only improves, detection performance for drone-acquired images, but also significantly reduces the computational burden.

In addition to the standard comparison, an analysis of the impact of individual changes in the architecture on the effectiveness of the final solution was also conducted.

General remarks:

The article is written in a readable and understandable way, but the authors did not shy away from some faults, often debatable - for example, the article contains a lot of acronyms, especially regarding the names of modules and algorithms, they are not always properly, it seems to me that in this type of work a list of acronyms would make it easier to read. The introduction is written correctly, but the description of existing solutions could be expanded a bit.

Part of the text in Section 1 (Introduction), is later repeated in a slightly altered form. Section 2.1 briefly describes general object detection algorithms, but the authors forget that there are also methods that do not use deep learning techniques. 

Detailed remarks:

Detailed comments can be found as comments in the uploaded pdf file.

Reviewer 2 Report

Minor style editing required

Reviewer 3 Report

-> You mention resource limitations with drones, but this is not elaborated upon. What specific limitations are you addressing, and how do these limitations affect object detection?

-> Explain why YOLOv7 was chosen as the base model. What advantages does it offer over other object detection architectures for drone imagery?

-> Clarify the rationale behind specific modifications, such as the removal of the P5 detection head and the introduction of the P2 detection head. How do these changes improve small object detection?

-> Provide more technical details about the CBS module, TGM-CESC module, and Head Context-Enhanced Method (HCEM). How do these modules work, and why were they incorporated into the model?

-> Explain the concept of pixel-level constrained sparse convolution masks and their role in the model. How does this contribute to computational efficiency?

-> Discuss the reasons for selecting the VisDrone and UAVDT datasets for evaluation. What are the characteristics of these datasets, and how do they represent real-world scenarios?

-> Provide concrete evidence or examples from the evaluation to support claims of robustness and reliability in aerial image detection.

-> Clearly state the unique contributions of Efficient YOLOv7-Drone to the field of object detection in drone aerial imagery. What sets this model apart from existing approaches?

Extensive English improvements are required like reducing the long lines.

Round 2

Reviewer 2 Report

In the captions to figure 5 and 6, the terms deeper and lighter do not explicitly describe the palette used. A scale bar with the color palette would really be helpful.

A couple of punctuation errors in and around the new (red) material.
